# Novel Nano-Drug Delivery System for Brain Tumor Treatment

**DOI:** 10.3390/cells11233761

**Published:** 2022-11-24

**Authors:** Ziyi Qiu, Zhenhua Yu, Ting Xu, Liuyou Wang, Nanxin Meng, Huawei Jin, Bingzhe Xu

**Affiliations:** 1School of Biomedical Engineering, Shenzhen Campus of Sun Yat-Sen University, Shenzhen 518107, China; 2School of Biomedical Engineering, Sun Yat-Sen University, Guangzhou 510275, China; 3Sun Yat-Sen University First Affiliated Hospital, Guangzhou 510060, China

**Keywords:** glioma, brain tumor, blood–brain barrier, nano-drug delivery systems

## Abstract

As the most dangerous tumors, brain tumors are usually treated with surgical removal, radiation therapy, and chemotherapy. However, due to the aggressive growth of gliomas and their resistance to conventional chemoradiotherapy, it is difficult to cure brain tumors by conventional means. In addition, the higher dose requirement of chemotherapeutic drugs caused by the blood–brain barrier (BBB) and the untargeted nature of the drug inevitably leads to low efficacy and systemic toxicity of chemotherapy. In recent years, nanodrug carriers have attracted extensive attention because of their superior drug transport capacity and easy-to-control properties. This review systematically summarizes the major strategies of novel nano-drug delivery systems for the treatment of brain tumors in recent years that cross the BBB and enhance brain targeting, and compares the advantages and disadvantages of several strategies.

## 1. Introduction

Brain tumors can be classified into two major classes: primary brain tumors that start in the brain and secondary brain tumors that are generated by the cancer cells that migrated from other parts of the body [1]. Glioma is the most common primary brain tumor, accounting for about 80% of all cases [2,3,4]. The World Health Organization (WTO) classifies gliomas into three categories: astrocytoma, oligoastrocytoma, and oligodendroglioma [5]. Astrocytoma can be further classified into grades I to IV, with grade IV being the most aggressive form of glioblastoma multiforme (GBM) [1,5,6]. Unfortunately, most GBM are aggressive and destructive with a 2-year overall survival rate of 25% after standard treatment [3]. One of the reasons for the poor prognosis of gliomas is the lack of brain tumor-targeted drug delivery systems capable of crossing the blood–brain barrier (BBB) and the blood–brain tumor barrier (BBTB).

The BBB refers to the barrier between plasma and brain cells, consisting of brain capillary walls and glial cells (Figure 1a) [7]. The BBB selectively penetrates ions, nutrients, and small molecules, prevents large molecules and unwanted cells from entering the brain, and contains a very rich efflux pump that discharges toxic substances, metabolites, and other waste out of the brain [2,3]. Tight junctions on the BBB exclude large hydrophilic molecules, such as proteins and peptides, unless they can be transferred via receptor- or adsorption-mediated endocytosis [8]. However, the endothelium/transcytosis activity of the brain endothelium is much lower compared to the peripheral endothelium. It has been reported that 98% of small molecules and 100% of large molecules cannot penetrate across the BBB [9]. Only some gas molecules or tiny, highly lipid-soluble molecules can enter the brain through paracellular pathways [10]. Therefore, the BBB can protect the brain from toxic substances in the blood and is a necessary prerequisite to ensure the normal function of brain neurons [3]. However, these efflux transporters not only protect the brain but also prevent many antitumor drugs from entering the brain through the BBB, thereby reducing the effectiveness of the drugs. In brain tumors, with the development of the tumor, the structure and function of the BBB are damaged to varying degrees, and new blood vessels–the BBTB–are gradually formed [5,10] (Figure 1b). In low-grade gliomas, the BBTB is similar in structure and function to the normal BBB, whereas in high-grade gliomas, increased permeability could be identified on the BBTB [3]. Although some drugs can cross the blood–brain barrier, such as TMZ, paclitaxel, doxorubicin, etc., their efficacy and ability to cross are limited [6] and high dose are often required. A nano-drug delivery system with easily controllable properties such as size, shape, and surface properties is an ideal answer to the BBB issues [11,12]. Through the design of these properties, the pharmacokinetic properties, BBB crossing ability, and brain tumor targeting ability of drug carriers can be controlled to achieve the best therapeutic effect. In this review, we will highlight some strategies to enhance the BBB traversal capability and brain tumor targeting of nanodrug delivery systems, and analyze and discuss the advantages and disadvantages of these strategies.

## 2. Strategies for Passing through the Blood–Brain Barrier

### 2.1. Surface Modification for Crossing BBB

There are many transport mechanisms for substances passing through BBB, such as paracellular pathways, receptor-mediated transport, transporter-mediated transport, adsorption-mediated transport, and cell-penetrating peptides, etc. [13,14,15,16,17] (Figure 2). Nano-drug carriers have easily controllable surface properties, and the corresponding BBB crossing ability can be easily controlled through different surface modifications (Figure 3a). In this section, several commonly used surface modification strategies will be introduced according to crossing mechanisms.

#### 2.1.1. BBB Receptor Targeted Ligand

Receptor-mediated endocytosis (RMT) nanodrug delivery systems are one of the most mature brain-targeting strategies with high specificity and selectivity. Some receptors are highly expressed in the blood–brain barrier, brain tumor cells, or related blood vessels, but have low or no expression in normal tissues, such as the transferrin receptor (TRs), scavenger receptors (SRs), insulin receptors (IRs) and lipoprotein receptors (LPRs) [18]. The corresponding ligands can be modified on drug carriers to guide them across the blood–brain barrier, thereby enhancing drug efficacy. Upon the binding of ligands on their corresponding receptors on the BBB, ligands and ligand-decorated molecules/carriers undergo endocytosis via membrane invagination forming intracellular transport vesicles [15].

Related studies have shown that the transferrin receptor (TR) is generally expressed in a variety of normal tissues, but the expression level is low in most tissues, while it is highly expressed in brain capillary endothelial cells forming the BBB [15]. Suma Prabhu et al. engineered superparamagnetic iron oxide nanoparticles (SPION) based polymeric nanocomposites which permeate the BBB by tagging transferrin/polysorbate-80 [4], and Sajini D et al. coupled carbon dots (C-dots) with the targeted ligand transferrin and two anticancer drugs–epirubicin and temozolomide–and constructed a triple coupling system with an average particle size of only up to 3.5 nm [19]. The results of these two studies show that the drug delivery system modified with transferrin ligand has stronger blood–brain barrier permeability and better therapeutic effect than the unmodified system.

Angiogenesis plays an important role in tumor growth, among which vascular endothelial growth factor and angiogenin family are the main angiogenic factors in GBM [20]. Anti-angiogenesis methods that control angiogenesis at various stages can prevent the development of tumors and can be used as a complementary method for cancer treatment. NRP-1, a transmembrane glycoprotein, is a co-receptor for members of the vascular endothelial growth factor (VEGF) family, expressed on neovascular endothelial cells, and plays an important role in promoting tumor angiogenesis [21]. Therefore, targeting the NRP-1 receptor could enhance the permeability of the drug transport system to the BBB at the tumor. Several studies have used tumor homing and penetrating peptide (tLyp-1) or peptide iRGD to modify drug carriers to target NRP-1 receptors and enter the BBB via receptor-mediated endocytosis [14,22]. Lin Zhao et al. designed a bifunctional fusion polypeptide Tat-C-RP7 with both NRP-1 targeting and anti-angiogenic activities, which have an excellent ability to penetrate the BBB and anti-tumor angiogenesis for the treatment of gliomas [23]. In this method, RP7 peptide binds to the overexpression of VEGF receptor on brain tumors to inhibit the phosphorylation of VEGFR2, thereby affecting the VEGFR2 signal pathway and inhibiting tumor angiogenesis. However, several receptors like integrin, folate, transferrin, etc., are also expressed in most tissues, which limits the brain-specific delivery [24]. Therefore, the development of highly specific brain-targeting ligands has attracted more and more attention in recent years. Tao Sun et al. used substance P that can target neurokinin-1(NK-1) receptor which is vastly distributed among the CNS system to connect with drug delivery system to improve brain targeting of drug delivery system [25], and Meenu Vasudevan et al. used TGN peptide with superior brain targeting efficacy as the targeting part and conjugated it to their glycopolymer (SAGC) to produce a hybrid, peptide decorated nanomicelle named “TSAGC” [24].

The shortcomings of receptor-mediated nanodrug delivery systems also limit their clinical applications. First, some receptors are highly expressed in other tissues of the human body, which can cause safety hazards. Second, brain targeting efficiency is reduced due to competition between endogenous substances and targeting ligands. Third, transport efficiency through the BBB is also limited by the poor number and permeability of receptors exposed to brain endothelial cells (BECs).

#### 2.1.2. BBB Transporter Targeted Ligands

A variety of specific transporters has been identified on cerebral vascular endothelial cells, which are expressed at higher-than-normal levels in selective cell types under pathological conditions [26], and help amino acids, carbohydrates, and other nutrients enter brain tissue along a concentration gradient. Under pathological conditions, these specific transporters are expressed at higher-than-normal levels in select cell types. Therefore, we can modify the surface of the nano-drug delivery system with amino acid analogs, carbohydrate analogs, or other substances that can be transported by transporters and cross the blood–brain barrier through the transporter system. Several studies have established GLUT1 as a validated target for the transporter-mediated transcytosis of nanoparticles. Studies have shown that nano-drug delivery systems modified with glucose transporter substrates can cross the BBB through transporter-mediated endocytosis [27]. Hao Wu et al. developed a polymer nano delivery system targeting the BBB through surface maltobionic acid (MA), in which MA is a glucose derivative that can be recognized by GLUT1 and then trigger the GLUT1-mediated BBB/BBTB transcytosis [28]. L-type amino acid transporter 1 (LAT1) is one of the most widely studied transporters for drug delivery across biological barriers [16]. LAT1, which is overexpressed on both the BBB and glioma cells, was selected as a target and modified glutamate-d-α-tocopherol polyethylene glycol 1000 succinate copolymer (Glu-TPGS) in liposomes to enhance the BBB penetration and glioma therapy [29]. Another transporter commonly used to deliver drugs across the BBB is choline transporter. Li et al. chose a choline derivate with high choline transporter affinity as the BBB and glioma dual targeting ligand [30]. The results of in vitro and in vivo experiments showed that the drug delivery system modified with choline bioderivatives could induce more apoptosis of glioma cells compared with the unmodified drug delivery system.

Transporters have certain advantages as targets for drug delivery. The substances they transport are usually small molecules with little immunogenicity [26] and a wide range of selectivity. Therefore, one ligand can target multiple transporters, thereby increasing the transport efficiency. In addition, transporters are highly expressed in tumor sites, and their ligands have certain specificity. The disadvantage of transporters is that their ligand specificity is not particularly high, which may lead to the off-target transport of drugs and cytotoxicity.

#### 2.1.3. Substances That Trigger Adsorption of the BBB

Many brain drug delivery strategies have focused on adsorption-mediated endocytosis (AMT), which is triggered by electrostatic interactions between cationic molecules and anionic microdomains on the cytoplasmic membrane of brain capillary endothelial cells [17]. Therefore, positively charged protein or polypeptide modified nanodrug delivery systems can trigger adsorption mediated endocytosis. Feng Xu et al. prepared cationic bovine serum albumin (CBSA) conjugated nanoparticles ( CBSA-NP) and proved through experiments that they can be effectively absorbed by brain tissue, but did not lead to a general enhanced uptake into all tissues [31]. Recently, various peptides have been used to modify nano drug delivery systems so that drugs can be transported to brain tumors through brain-targeted transport mediated by adsorption. Sushant et al. designed dual functionalized liposomes by modifying their surface with transferrin and a cell penetrating peptide (CPP) for receptor and adsorptive mediated transcytosis, respectively [32]. Similarly, Shi et al. proposed a CPP/Ab bifunctional thermosensitive liposome DDS for the treatment of GBM [33]. Additionally, Neves et al. found and proved that novel peptides derived from dengue virus capsid protein, especially a peptide named peph3, are a very good candidate to be used as a peptide shuttle, taking cargo in and out of the brain [34]. Cationic proteins and antibodies are good brain-targeting agents and have optimized pharmacokinetic characteristics, but their toxicity and immunogenicity are the key problems to be overcome in drug development in the future [17].

### 2.2. Biomimetic Nano Drug Delivery System to Cheat the BBB

Recently, biomimetic nano-drug delivery systems (DDS) have received extensive attention due to their advantages in drug delivery. Biomimetic DDS, a new type of DDS developed by directly exploiting or imitating complex biological structures and processes, has been studied for the treatment of a variety of diseases [35]. Specifically, the term nano biomimetic drug carrier refers to nano particles extracted from organisms, including cells, cell membranes, extracellular vesicles, virus carriers, and endogenous proteins (Figure 3b). These biological vectors inherit structural and functional complexity from original donors, serving as native substances to reduce undesired immune response and avoid being cleared immediately [36]. Therefore, bionic DDS may have the advantages of penetrating the BBB, good biocompatibility, optimal accumulation in the targeted area, protecting drugs from degradation, and so on [35]. This section will introduce bionic drug carriers according to their different sources.

#### 2.2.1. Cell-Mediated Drug Delivery System

Some endogenous cells have the inherent ability to cross the blood–brain barrier and penetrate tumor sites, so the use of endogenous cell-mediated drug delivery systems is a potential brain-targeting strategy (Figure 2f). Cell-mediated drug delivery systems use specific cells as drug carriers to deliver drugs to target sites [37]. Immunocytes, mononuclear phagocytes (MP: monocytes, macrophages, and dendritic cells), lymphocytes, and neutrophils, as well as stem cells exhibit an intrinsic homing property enabling them to migrate to sites of injury, inflammation, and tumor across the BBB [37,38,39]. Thus, cell mediated drug-delivery targeting systems have received considerable attention for their enhanced therapeutic specificity and efficacy in the treatment of the disease [40]. Cell-mediated drug delivery systems have also received extensive attention and applications in the treatment of brain tumors. Neutrophils are white blood cells that are naturally chemotactic for inflammatory signals and can freely pass through vein walls and body tissues [40]. Xue et al. report the application of NEs that carry paclitaxel (PTX)-loaded liposomes to suppress postoperative glioma recurrence [41]. They used neutrophils as drug carriers to improve the brain targeting of drugs. Dendritic cells (DCs) and macrophages as delivery vehicles can effectively overcome the BBB and other structural and metabolic barriers that impede drug entry into the GBM [40,42]. Based on their previous research, Yu et al. prepared a DC-mediated doxorubicin polyglycerol nano diamond composite (nano DOX) delivery system for inducing an enhanced DC-driven anti GBM immune response [42]. Hao et al. synthesized stem cell-mediated delivery of nanogels loaded with ultrasmall Fe^3^O^4^ NPs for enhanced MR imaging of tumors [43].

#### 2.2.2. Cell Membrane Drug Delivery System

Synthetic nanoparticles that coat native cell membranes can be prepared by harvesting intact cell membranes from native cells and coating them on nanoparticle surfaces [44,45]. As a new generation of biomimetic nanoparticles, cell membrane-coated nanoparticles combine the complex biological functions of natural cell membranes with the physicochemical properties of synthetic nanomaterials for more efficient drug delivery [44,45,46]. Membrane-coated nanoparticles (NPs) possess unique properties such as immune escape, long blood circulation time, specific molecular recognition, and cellular targeting [47]. The cell membrane sources used for the synthesis of composite nanoparticles mainly include red blood cells, platelets, white blood cells, cancer cell membranes, and stem cells [48].

Red blood cell membrane-coated nanoparticles (RBCNPs) are widely used in the treatment of brain tumors. RBCNPs often enhance their brain targeting or brain tumor targeting through surface modifications [49,50]. Many cancer cells inherently have self-targeting abilities to adhere to their homologous cells, which is the so-called “homologous adhesion” [48]. In recent years, the application of cancer cell membrane-coated nano drug delivery system in brain tumor treatment has gradually increased due to its excellent brain targeting and brain tumor targeting characteristics. For example, Pasquale et al. designed and prepared an innovative nanoplatform consisting of Dox-loaded BNNTs coated with GBM cell membrane (DoxCM-BNNTs) [51]. Based on the homotypic recognition of GBM cells, the drug delivery system has improved its targeting performance. In addition to using homotypic cell membrane, Wang et al. also proposed a new strategy to construct biomimetic nanocarriers of brain metastatic tumor cell membrane inspired by brain metastatic tumor [52]. This is the first trial to camouflage nanoparticles with brain metastatic tumor cell membranes to traverse the BBB for the therapy of brain tumors.

#### 2.2.3. Extracellular Vesicle Drug Delivery System

Extracellular vehicles (EVs), including exosomes, microvesicles, and apoptotic bodies, are nanosized membrane vesicles derived from most cell types [53,54]. EVs are potent biological vectors capable of transporting functional biomolecules between cells over large intercellular distances [55]. EVs are considered important regulators of crosstalk between neurons, astrocytes, microglia, and oligodendrocytes in both physiological and pathological states of the central nervous system (CNS) [54]. Due to its natural source and functional requirements, EVs have high biocompatibility, high stability, limited immunogenicity, and inherent homing ability [53,54,56]. In addition, EVs have the ability to cross the BBB, and therefore they are highly potential drug transporters for the treatment of brain tumors [54].

Among several EV types, exosomes have received the most attention in recent decades. The small size of exosomes and the native protein and lipid components on the exosome membrane give them a natural cellular uptake capability and can effectively penetrate biological barriers including the BBB [57]. More recently, Niu and colleagues demonstrate a distinct design by synergistically combining natural grapefruit extracellular vehicles (EVs) and doxorubicin (DOX)-loaded heparin-based nanoparticles (DNs) to construct a biomimetic drug delivery system for highly efficient drug delivery in glioma treatment [58]. The biomimetic EV DNS has good brain targeting, brain tumor targeting and drug delivery capabilities, providing a new drug delivery platform for the treatment of brain tumors. Zhu et al. used exosomes produced by endogenous embryonic stem cells as drug carriers. In this study, the anti-GBM effect of ESC-exos is confirmed and then c(RGDyK)-modified and paclitaxel (PTX)-loaded ESC-exos, named cRGD-Exo-PTX, are prepared [59]. Studies have confirmed that ESC-exos also have a certain anti-GBM effect compared with other drug carriers. Therefore, ESC-exos are powerful therapeutic vectors for the treatment of GBM.

#### 2.2.4. Virus Nano Drug Delivery System

Viruses naturally exhibit several characteristics that are of interest for drug delivery due to their intrinsic ability to avoid immune system recognition and enter cells to “deliver” their genes into a host for self-replication [60]. In recent 20 years, viral vectors have been used in gene therapy of brain tumors because of their ability to load genes [61]. However, these viral vectors generally do not have the ability to passively cross the BBB, and often need to bypass the BBB through other routes of administration [62]. In order to solve this problem, drug vectors with artificially designed virus capsid and new sources of virus vectors began to appear.

Adeno-associated virus (AVV) is one of the most promising gene delivery tools for the treatment of brain diseases, because its viral capsid has a high natural affinity with neurons [62,63,64]. AAV-mediated therapeutic agents for diseases of the CNS were initially developed to be used via local injection due to the presence of the BBB [64]. BBB-crossing AAV vectors were developed by manually designing AAV capsids [64]. For example, Matheus et al. used AAV9, a variant of AAV, as the vector encoding the therapeutic gene to achieve the treatment of brain tumors [65]. Unlike AAV, AAV9 can pass through the BBB after intravenous injection [65]. Moreover, plant virus is an attractive chemotherapeutic drug carrier. Since plant viruses are always nonintegrated and non-replicating in mammalian systems, they can provide security advantages [66]. Patricia et al. investigate the use of CPMV for the delivery of MTO to treat GBM, while the CPMV is a plant virus that can be rapidly internalized by BBB cells [64,67].

#### 2.2.5. Protein-Based Drug Delivery System

Proteins and peptides have many biological roles in the brain, such as controlling the internal environment of the brain, including cerebral blood flow, permeability of the BBB to nutrient supply, neurotransmission, neuromodulation, and various roles in the immune system [5], which offers the potential for use as a viable drug carrier for brain-targeted drug delivery systems. In addition, the protein-based drug delivery system also has the advantage of biodegradability, biocompatibility, no or low toxicity, and ease of modification [68,69]. Protein nanocarriers are now drawing great interest as drug delivery systems targeting brain tumors [5]. For example, Heng et al. enhanced brain targeting by adding cations or mannose to the surface of albumin nanoparticles [70] while Lu et al. modified folate on the surface of albumin particles [71]. The biodegradable, nonantigenic, and non-toxic characteristics of human serum albumin (has) make it an ideal candidate for tumor targeting [69,72]. However, since albumin is expressed at very low levels in normal BBB vessels, passage of native albumin is difficult [5], and surface modification methods must be combined to enhance its ability to pass through the BBB. Ferritin and lipoprotein not only have good biocompatibility, immunogenicity, and biodegradability of natural proteins, but also exhibit natural targeting properties that albumin does not have [69]. Luciana et al. demonstrated the possibility of using a ferritin-based drug delivery system to deliver toxic molecules to brain tumors, which relies on the ability of H-ferritin (HFT) to carry multiple metals and cross the blood–brain barrier via an endocytic mechanism [73]. Lin Huang et al. demonstrated that E3 recombinant high-density lipoprotein (ApoE-rHDL) is a highly efficient nanoplatform with blood–brain barrier (BBB) permeability and used it to deliver siRNA to glioblastoma cells for the treatment of brain tumors [74].

### 2.3. Change the Mode of Administration to Bypass the BBB

Most of the time, the drug and drug carrier are administered orally and intravenously. While tailored nanodrug carriers can gain the ability to target and cross the blood–brain barrier, these drugs need to pass through the systemic circulation, and the drug will always reach other organs, leading to toxicity–especially when patients are severely ill and require large doses [75]. Therefore, by changing the drug delivery strategy, such as the use of local administration, it is possible to bypass the BBB and deliver drugs directly in the brain (Figure 3c).

#### 2.3.1. Local Delivery

The majority of the nanoparticles targeting the brain are administered either intravenously or orally, resulting in some amount of drug reaching other organs and leading to toxicity [75]. Local administration to the GBM has several advantages over systemic administration, such as bypassing the BBB and increasing the bioavailability of the therapeutic agent at the tumor site without causing systemic toxicity [76,77]. Local drug delivery is achieved by injecting a drug or drug carrier into an intracranial inoperable tumor or tumor resection cavity as the chosen drug is delivered using injectable and/or implantable systems and could target infiltrating cells which are not accessible at tumor resection [78]. Biomaterials for local delivery of chemotherapy can be divided into three main categories: implantable polymeric drug delivery devices (e.g., a degradable wafer), hydrogel-based delivery systems, and injectable polymeric-based delivery systems [76,79]. For surgically resected brain tumors, it can be treated by implantable polymer delivery devices and hydrogel delivery systems, but for unresectable malignant brain tumors, the injectable polymer delivery system can only be administered.

Gliadel, a carmustine-filled polymer wafer, is currently the only local delivery device that has been approved by the FDA for use in GBM [76,78,79]. The wafer needs to be surgically implanted and is inherently rigid, so there is a risk of infection and inflammation. The hydrogel drug delivery system has good biocompatibility and mechanical properties and can match the surrounding tissues. They can not only be implanted into the brain through surgery, but also by direct injection into the brain, making it easier to deliver drugs based on the hydrogel delivery system [76]. Bastiancich et al. developed an injectable gel-like nano delivery system consisting of lipid nanocapsules loaded with anticancer prodrug lauroyl-gemcitabine (GemC12-LNC) in order to obtain a sustained and local delivery of this drug in the brain [80]. Studies have shown that the GemC12-LNC hydrogel nanoparticle delivery system is safe and effective in the treatment of GBM. Injectable polymer delivery systems in local drug delivery systems are usually administered by stereotactic injection of injectable polymers into tumors or by convection-enhanced delivery (CED). Limited penetration of drugs in solid tumors is a major cause of poor therapeutic indices of many chemotherapeutic agents [79]. Different from the diffusion-based method mentioned above, CED utilizes a continuous pressure gradient to drive bulk flow of agents which are infused directly into the tumor resection cavity, enabling large distribution of high drug concentrations, while avoiding systemic toxicity [77,81]. Furthermore, CED combined with nano carrier can improve the poor permeability and retention of drugs in tumors caused by injection of free drugs [76]. Singleton et al. invented a novel panobinostat water-soluble nano micelle formulation administered with CED [82]. The local delivery scheme overcomes the disadvantage that panobinostat cannot cross the BBB, gives full play to the efficacy of the drug, and effectively prolongs the survival time of animals with gliomas.

#### 2.3.2. Nose-to-Brain Delivery

Topical administration is generally used when the patient is very ill because it is invasive and inconvenient compared to other modalities. Intranasal administration can bypass the BBB and directly deliver drugs to the brain through the olfactory area or the trigeminal nerve pathway, thereby increasing drug bioavailability and efficiency [83,84,85]. In addition to these advantages, intranasal administration also has the advantages of non-invasiveness, avoidance of systemic toxicity, rapid onset of action, high clearance rate, and so on [84,86]. However, intranasal administration also has some problems, such as limited drug volume (max: 150 μL/nostril), the enzymatic degradation of drugs, limited drug absorption due to mucociliary clearance, and low drug retention at the absorption site [83,84].

The main trend of nose-brain drug delivery systems is to design formulas with increased permeability, low clearance, and high mucus adhesion for nasal brain drug delivery [86]. Edilson Ribeiro de Oliveira junior et al. described a biodegradable and biocompatible polycaprolactone nanoparticle for melatonin (MLT) delivery from nose to brain to treat GBM [87]. In this study, MLT was encapsulated by nano carrier PCL with high encapsulation efficiency, which overcomes the disadvantage of drug degradation in nasal brain administration. Mucociliary clearance is a major obstacle to drug delivery. The use of adhesives and tackifiers is expected to increase the residence time of drugs in the nasal cavity for better absorption [83]. Chitosan is a kind of viscous polymer, which can increase the retention time of drugs at the absorption site and improve the absorption efficiency of drugs. Nat á LIA n. Ferreira et al. designed adhesive nanoparticles (NPS) based on polylactic acid glycolic acid (PLGA) and oligochitosan (OCs) for co-delivering alpha-cyano-4-hydroxycinnamic acid (CHC) and the monoclonal antibody cetuximab (CTX) to the brain [88]. In addition to these targeted strategies, liposome drug delivery system has attracted much attention in nose-brain drug delivery because of its adhesion ability and nasal application potential [89]. In the current research, liposome nanocarriers are considered to have the best biocompatibility and the least toxicity among many nanocarriers [86]. Rohini G et al. designed a stable nanostructured lipid carrier (NLC) loaded with curcumin (CRM) and used nose-brain administration to treat brain tumors [90]. The results showed that after nasal administration of CRM-NLC, the concentration of curcumin in the brain increased significantly, and the cytotoxicity to tumor cell astrocytoma glioblastoma (U373MG) was significantly higher than that of free CRM. This study also indicates that nasal CRM-NLC nano drug delivery system is a promising anti-brain-tumor drug delivery system.

### 2.4. Destroy the BBB by Physicochemical Methods

Several physical and chemical methods can temporarily disrupt the blood–brain barrier and alter its permeability. Intra-arterial injection of hyperosmotic agents such as mannitol or alkylglycerol can reversibly disrupt the BBB [6]. Relevant research shows that the permeability of the BBB is highly temperature-dependent, allowing significantly increased transport of substances at temperatures above 38–39 °C [91]. Drug-loaded magnetic nanoparticles generate heat in an alternating magnetic field, which can destroy the BBB and increase the ability to pass through the BBB [92]. In addition, the BBB can also be opened temporarily through the combination of focused ultrasound and microbubbles (Figure 3d). Focused ultrasound combined with microbubbles is a commonly used technology, which will be introduced in this section.

#### Focused Ultrasound Temporarily Destroys the BBB

Focused ultrasound (FUS) can help drugs or drug carriers enter the brain by opening the local BBB briefly and reversibly. Although the application of FUS alone can increase the permeability of the BBB, the ultrasonic intensity required in this case is close to the ablation range of tissue [93]. This may cause damage to brain tissue. FUS combined with microbubbles can open the BBB at low sound intensity, which makes the use of FUS safer. Nathan et al. used FUS with microbubbles to repeatedly interfere with the BBB of the visual cortex and lateral geniculate nucleus of rhesus monkeys [94]. They found that the treatment did not impair the animals’ behavior and vision, and reliably and safely opened the blood–brain barrier without impairing brain tissue function. Moreover, the penetration of the BBB can return to baseline levels within 6–24 h [10,95]. At present, it is considered that the main mechanism of opening the BBB by FUS with microbubble is the cavitation effect produced by FUS sonication. Focused ultrasound (FUS) can cause the expansion of the gaseous microbubbles until they rupture in a process termed inertial cavitation, while lower FUS power levels instead lead to oscillation of the microbubbles, known as stable cavitation [10,95]. Stable cavitation causes microbubbles to oscillate and exert mechanical force on the surrounding vascular wall, which temporarily increases the permeability of the BBB without damaging the surrounding tissue [10,93,95,96]. At present, the technology of FUS combined with magnetic resonance imaging (MRI) is more commonly used, because it can accurately locate the pathological area and achieve more accurate administration.

For the treatment of brain tumors, one of the biggest challenges is the heterogeneity of blood vessels in tumor tissues, which leads to the uneven distribution of drugs [10]. With FUS, it is possible to increase the permeability of the BBB at a given location, overcoming this obstacle. FUS is usually combined with imaging examination to increase the accuracy of administration in which MRI is the most useful form for the diagnosis of GBM [97]. Magnetic resonance imaging guided focused ultrasound (MRgFUS) nanodrug delivery systems generally need to be used in combination with paramagnetic nanoparticles to increase imaging contrast. Daniel et al. investigated a nano drug delivery system using Cisplatin-conjugated gold nanoparticle (GNP-UP-Cis) combined with MRgFUS [98]. The results showed that MRgFUS can increase the permeability of the BBB and the accumulation of drugs in tumor sites, and greatly inhibit the growth of GBM tumors. The combination of MRgFUS and other brain-targeting strategies can further improve the efficiency of drug delivery. Chan et al. loaded FePt nanoparticles and adriamycin into nanobubbles, modified transferrin on the surface of nanobubbles, and combined it with high-intensity focused ultrasound (HIFU) for drug delivery [99]. This study further enhances the accumulation of drugs in tumors by combining focused ultrasound with receptor-mediated brain targeting technology. Wu et al. modified Cu^2^ − xSe nanoparticles on the surface of hollow mesoporous organic silica nanoparticles through a disulfide bond to construct a new drug-loading system [97].

## 3. Strategies for Enhanced Brain Tumor Cells Targeting

To reduce the damage of nanomedicines to normal brain function and immune system, the targeting of brain nanodrug delivery systems to brain tumor regions should be enhanced. Currently, brain tumor targeting mainly relies on active targeting strategies, which enhance the accumulation of drugs at tumor sites through passive targeting strategies. Several emerging strategies have recently emerged, such as magnetically targeted nano-drug delivery systems, tumor microenvironment-triggered drug delivery systems, etc. [97,100]. This section will focus on passive targeting, active targeting, and magnetic targeting.

### 3.1. Passive Targeting

Due to the highly leaky vasculature environment around the tumor, passive targeting systems have been applied to the design of brain cancer-targeting nanomedicines [18]. Passive targeting of brain tumors mainly relies on enhanced permeability and retention (EPR) effects. In the pathological state of brain tumors, the BBB may be disturbed to some extent by edema, swelling, and increased pressure in the brain [6]. Nanoparticles of a certain size or smaller can enter the tumor through the gap between endothelial cells, which is the EPR effect caused by the collapse of blood vessels caused by the formation of solid tumors [5], resulting in a higher accumulation of nanoparticles in brain tumors. The success of the ERP effect prompted researchers to develop nanoparticles with different physicochemical properties, including size, surface charge, surface hydrophilicity, and geometry, in order to enhance the aggregation of drugs [44]. Relevant studies have shown that the nanoparticle size range for ideal EPR effect is 10–200 nm [5,18]. When larger than 200 nm, the particles cannot fully penetrate the tumor vasculature and interstitial space, while when smaller than 10 nm, the kidneys clear them, resulting in the inability of particles to accumulate at the tumor site [5,101]. In addition, the lipophilicity and stability of nanoparticles is also an important factor affecting the ERP effect. Liposomes and polymer nanoparticles are often used to encapsulate drugs to enhance the ERP effect. PEGylation modification is widely used to modify the drug itself or drug carrier to increase the systemic circulation time of nanoparticles, so as to enhance the EPR effect [102]. The biomimetic nano drug loading system also has an enhanced EPR effect because of its long body circulation. While the EPR effect may be in effect for administered nanoparticles, the majority (>95%) of administered nanoparticles are known to accumulate in other organs, in particular the liver, spleen, and lungs [102]. Nano drug delivery systems do not rely on passive targeting strategy alone but are combined with active targeting strategy.

### 3.2. Active Targeting

Active targeting usually refers to targeting by ligand-receptor-specific interactions between a drug or drug carrier and the target cell [102]. Compared with passive targeting, active targeting is selective for brain tumors and can effectively deliver therapeutic drugs or diagnostic reagents to the lesion site and reduce cytotoxicity [18]. Some receptors are closely related to tumor growth. They are highly expressed in blood vessels near brain tumors or in brain tumor cells, but not expressed or expressed low in other tissues. Therefore, by modifying the corresponding ligands on the surface of drugs or drug carriers, the receptors can guide them to aggregate at brain tumors or enter tumor cells to exert therapeutic effects. In the introduction of brain targeting strategy, this paper also mentioned the receptor-mediated targeting strategy, but these receptors are expressed in the BBB and brain tumors, which increases the brain targeting of drugs; however, they lack the targeting of brain tumors in the brain [103], which may lead to severe neurotoxicity.

In order to solve this problem, receptor-mediated endocytosis with high expression in tumor sites and low expression in the BBB are selected to enter brain tumors, such as NRP-1 receptor [14,22], integrin receptor [59], interleukin receptor, and so on [104]. At present, the development of dual targeted drug delivery system provides a more safe and effective method for brain tumor drug delivery. Dual-targeted drug delivery systems are two ligands that simultaneously modify the surface of the drug delivery system: one modifies the BBB and the other modifies the brain tumor [103] (Figure 4a). Therefore, the dual-targeted drug delivery system can improve drug efficacy and reduce toxicity and side effects to the body. TGN peptide (TGNYKALHPHNG) is selected from the 12-peptide library through phage display in vivo, which has a good brain targeting effect, while RGD peptide has been proved by many studies to have good tissue cell penetration and tumor targeting [22,24,105,106,107]. Shi et al. developed a glioma drug delivery system with iRGD/TGN double modified poly (amidoamine) dendrimer (PAMAM) encapsulated ATO. The experimental results show that iRGD/TGN-PEG-PAMAM-ATO has enhanced BBB penetration and GBM targeting, which effectively improves the efficacy of ATO, prolongs the medium survival time of mice, and reduces the systemic toxicity. This suggests that the dual targeting system is a promising technology in the treatment of glioma.

### 3.3. Magnetic Targeting

Magnetic targeted drug delivery is a method of using external magnetic field to manipulate magnetic drug carriers in vivo to reach tumor targets [108]. As magnetic targeting depends on the magnetic force between the external magnetic field and nanoparticles, the nano carrier used must be magnetic–that is, magnetic nanoparticles. Magnetic nanoparticles such as iron oxide nanoparticles (IONPs), superparamagnetic iron oxide nanoparticles (SPIONs), and fluorescent magnetic nanoparticles (MNPs) are widely used as diagnostic imaging agents and therapeutic carriers [109]. By coupling magnetic nanoparticles with specific ligands and guiding them by a magnetic targeting system, drugs can be targeted to brain tumor therapy. The magnetic targeting strategy is to expose MNPs to the external magnetic field by applying an external magnetic field to the head tumor after intravenous administration, which increases the movement of MNPs in the systemic circulation to the brain tumor [109,110] (Figure 4b). In recent years, many nano drug delivery systems using magnetic targeting strategy have been developed to treat brain tumors. Cui et al. developed a dual-targeting strategy by a combination of magnetic guidance and transferrin receptor-binding peptide T7-mediated active targeting delivery [111]. This study showed that compared with non-targeted NPs, this strategy increased cell uptake by more than 10 times and brain transmission by more than five times. The experimental results also showed that under the action of a magnetic field, the system improved the drug delivery efficiency, reduced adverse reactions, and improved the survival rate of mice with glioma in situ. Although the external magnetic field can gather the drug carrier near the brain tumor, it is difficult to accurately identify the brain tumor only by magnetic field guidance due to the lack of selectivity. Therefore, Lu et al. developed a dual thermal sensitive magnetic liposome (TML) with a thermal response and magnetic response to recognize the overexpressed epidermal growth factor receptor on the surface of cancer cells by conjugating with cetuximab (CET) [100]. Many studies have shown that the magnetic targeting strategy has the advantages of improving the curative effect, reducing drug dosage, and reducing side effects [100,110,111,112].

## 4. Conclusions

To date, glioma remains a difficult disease to cure completely because of its aggressiveness and poor prognosis. Chemotherapy is an irreplaceable method for clinical treatment of brain tumors, however, the BBB/BBTB have become the main obstacles. Drug delivery systems based on nanotechnology can effectively improve the drug delivery efficiency and reduce the high toxic side effects caused by conventional chemotherapy. Adding contrast agent in the drug delivery system can also visualize the treatment process of brain tumors, which is helpful to achieve accurate drug delivery. However, due to the drug resistance of glioma cells, drug failure is likely to occur. In addition, some drug delivery systems compete with endogenous substances in the human body, resulting in a low efficiency of drug delivery. At the same time, in order to reduce the toxicity of the drug carrier itself, rapid degradation materials are often used in the design of nanoparticles, which will also lead to the degradation of drugs before they reach the action site. Therefore, the development of multifunctional and multi-targeted nano drug loading systems is the main trend at present. Some gene therapy and antiangiogenic therapy methods can be used as strategies against the drug resistance of glioma. A biomimetic nano drug system has the structural and functional complexity of the original biological donor, has the natural ability to pass through BBB, and has high biocompatibility, which can effectively protect the drug from degradation. It is a promising drug carrier. In addition, a variety of related technologies have been developed to enhance the targeting of brain tumors, such as modifying tumor-targeting ligands on the carrier surface and targeting brain tumors through magnetic fields. Among them, the dual targeting strategy of brain tumors can reduce physical toxicity and increase the accumulation of drugs in tumors, which is a very effective method worth developing. So far, there is still a broad space for the development of nano-drug delivery systems. The combination of multiple strategies and the development of multifunctional nanoparticles provide a more effective therapeutic strategy for the treatment of GBM.

## Figures and Tables

**Figure 1 cells-11-03761-f001:**
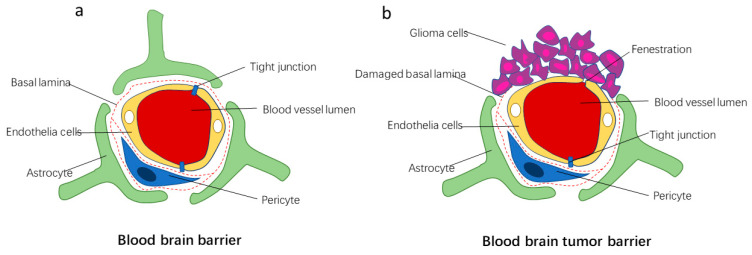
Schematic of composition and structure of blood–brain barrier and blood–tumor barrier.

**Figure 2 cells-11-03761-f002:**
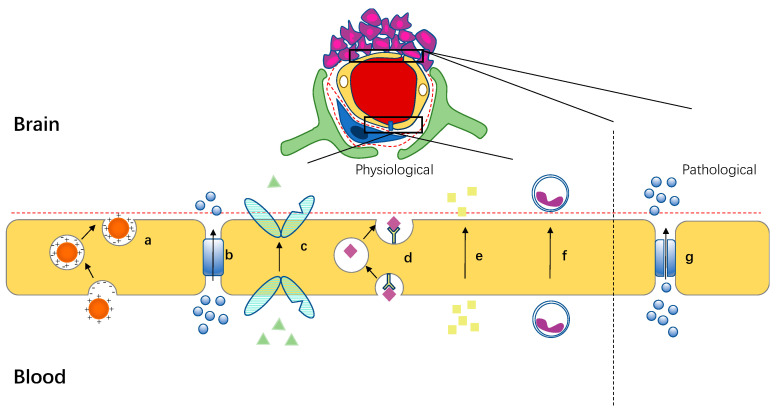
Schematic of various mechanisms across the BBB. (**a**) Adsorption-mediated endocytosis. (**b**) Water-soluble small molecules pass through the paracellular pathway. (**c**) Transporter-mediated endocytosis. (**d**) Receptor-mediated endocytosis. (**e**) Lipid-soluble small molecules pass through the transcellular pathway. (**f**) Cell-mediated endocytosis. (**g**) The tight junction of the BBB at the tumor is destroyed, which can increase the number of small molecules passing through the BBB.

**Figure 3 cells-11-03761-f003:**
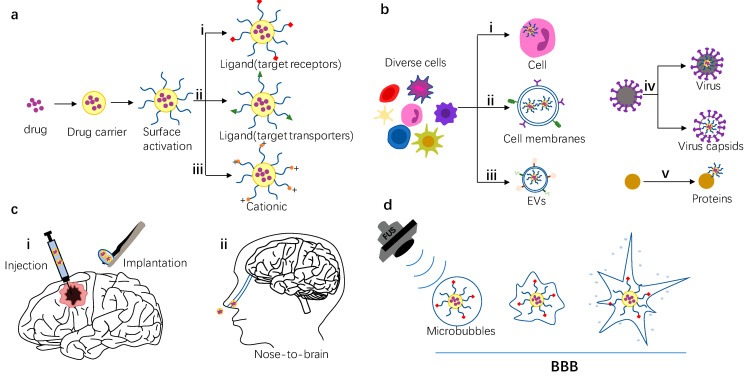
Schematic of strategy of nano drug carrier entering the brain. (**a**) Cross the BBB by surface modification. (i) Schematic of a nanomedicine carrier modified with ligands targeting receptors. (ii) Schematic of a nanomedicine carrier modified with ligands targeting transporters. (iii) Schematic of a nanomedicine carrier modified with cationic. (**b**) Biomimetic nano delivery system that can cheat the BBB. (i) Schematic of cells that can pass directly through the BBB, shown here as a neutrophil. (ii) Schematic of drug carrier coated with cell membrane. (iii) Schematic of extracellular vesicle drug delivery system. (iv) Schematic of virus nano drug delivery system. (v) Schematic of protein-based drug delivery system (**c**) Bypass the BBB by an unconventional route of administration. (i) Schematic of local administration mode, which can be realized by injection or surgical implantation. (ii) Schematic of nose-to-brain administration mode. (**d**) Schematic of temporary destruction of the blood–brain barrier by ultrasound combined with microbubbles.

**Figure 4 cells-11-03761-f004:**
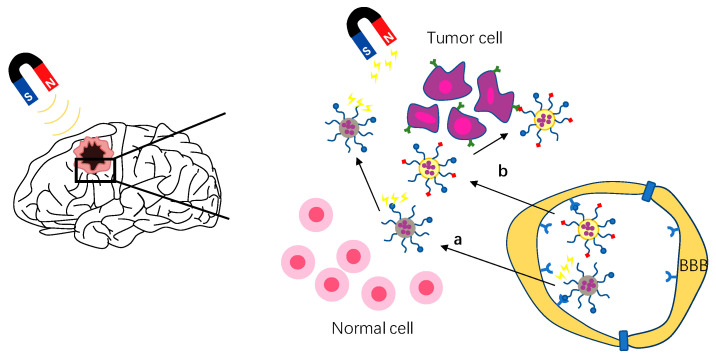
Schematic of strategies to enhance brain tumor targeting. (**a**) Schematic of magnetic targeting brain tumor strategy. (**b**) Schematic of active targeting strategy, shown here as double targeting strategy.

## Data Availability

This review did not report any data.

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
