# Peer review of "Novel Nano-Drug Delivery System for Brain Tumor Treatment"

_cells, 2022, doi:10.3390/cells11233761_

Round 1
Reviewer 1 Report
I read with much interest the manuscript entitled “Novel nano-drug delivery system for brain tumor treatment” By Ziyi Qiu, Zhenhua Yu, Ting Xu et al. It is a very interesting and well described review that describes novel therapeutic strategies in the treatment of glioma summarizing major approaches of novel nano-drug delivery systems. I think that Authors could also include the various phases of the angiogenesis, and specifically its representation as valid approach in tumor therapy. It can prevent the development of tumors and can act as a complementary therapy in cancer treatment. They can also describe the antiangiogenic approach which directed to a specific biomarker serves to inhibit or alter a specific pathway, through a link with specific membrane receptors overexpressed by glial cells, thus allowing the targeted release of the pharmacological compound.
In addition, they also can cite an interesting paper: “Nanoparticles drug‑delivery systems and antiangiogenic approaches in the treatment of gliomas” by Caffo M et al, Glioma 2018;1:183-8.
Author Response
Reviewer 1
I read with much interest the manuscript entitled “Novel nano-drug delivery system for brain tumor treatment” By Ziyi Qiu, Zhenhua Yu, Ting Xu et al. It is a very interesting and well described review that describes novel therapeutic strategies in the treatment of glioma summarizing major approaches of novel nano-drug delivery systems. I think that Authors could also include the various phases of the angiogenesis, and specifically its representation as valid approach in tumor therapy. It can prevent the development of tumors and can act as a complementary therapy in cancer treatment. They can also describe the antiangiogenic approach which directed to a specific biomarker serves to inhibit or alter a specific pathway, through a link with specific membrane receptors overexpressed by glial cells, thus allowing the targeted release of the pharmacological compound.
We thank the reviewer for the valuable comments. We have discussed the various phases of the angiogenesis in the revised version, which including the specific biomarker servers to inhibit or alter a specific pathway and a link with specific membrane receptors overexpressed by glial cells (page 4, lines 114-117, lines124-129).
In addition, they also can cite an interesting paper: “Nanoparticles drug‑delivery systems and antiangiogenic approaches in the treatment of gliomas” by Caffo M et al, Glioma 2018;1:183-8.
We thank the reviewer for the suggestions. We have cited the paper as ref. 20.
Reviewer 2 Report
I was pleased to review the article ID cells-2042168 entitled “Novel nano-drug delivery system for brain tumor treatment” for Cells. The review article summarizes the significant strategies of novel nano-drug delivery systems for treating brain tumors in recent years that cross BBB and enhance brain targeting, comparing the advantages and disadvantages of several methods.
The author claims in the abstract that the manuscript systematically reviews the topic, but the article resembles a narrative review. Please check the instruction for authors and PRISMA guidelines.
Please amend Figure 2: d) Receptor-mediated; e) Lipide-soluble.
Please check the whole text to avoid double citations throughout the entire manuscript, for example, in the sentence starting in Line 129: Tao Sun et al….. ends (23).
The conclusion section should be rewritten to provide better clarity and future perspectives in material development to improve brain tumor treatment.
Author Response
Reviewer 2
I was pleased to review the article ID cells-2042168 entitled “Novel nano-drug delivery system for brain tumor treatment” for Cells. The review article summarizes the significant strategies of novel nano-drug delivery systems for treating brain tumors in recent years that cross BBB and enhance brain targeting, comparing the advantages and disadvantages of several methods. The author claims in the abstract that the manuscript systematically reviews the topic, but the article resembles a narrative review. Please check the instruction for authors and PRISMA guidelines.
We appreciate the reviewer’s suggestions. We are sorry that our initial draft was not comprehensive enough, we have checked the PRISMA guidelines and made some modifications according to the reviewers' comments.
Please amend Figure 2: d) Receptor-mediated; e) Lipide-soluble.
We have amended Figure 2 as suggested (Page 3, line 78).
Please check the whole text to avoid double citations throughout the entire manuscript, for example, in the sentence starting in Line 129: Tao Sun et al….. ends (23).
We thank the reviewer for the suggestion. We double-checked references to avoid duplicate citations.
The conclusion section should be rewritten to provide better clarity and future perspectives in material development to improve brain tumor treatment.
We thank the reviewer for the comment. We have rewritten most of the conclusions to provide better clarity and future perspectives on materials development to improve brain tumor therapy (Page 13, lines 555-570).
Round 2
Reviewer 2 Report
The manuscript was improved and is suitable to be published in its current form.